# Formal enantioconvergent substitution of alkyl halides via catalytic asymmetric photoredox radical coupling

Jiangtao Li[1], Manman Kong[1], Baokun Qiao[1], Richmond Lee [2], Xiaowei Zhao[1] & Zhiyong Jiang [1]

Classic nucleophilic substitution reactions ($S_N1$ and $S_N2$) are not generally amenable to the enantioselective variants that use simple and racemic alkyl halide electrophiles. The merging of transition metal catalysis and radical chemistry with organometallic nucleophiles is a versatile method for addressing this limitation. Here, we report that visible light-driven catalytic asymmetric photoredox radical coupling can act as a complementary and generic strategy for the enantioconvergent formal substitution of alkyl haldies with readily available and bench-stable organic molecules. Single-electron reductive debrominations of racemic α-bromoketones generate achiral alkyl radicals that can participate in asymmetric $C_{sp3}$–$C_{sp3}$ bonds forming cross-coupling reactions with α-amino radicals derived from N-aryl amino acids. A wide range of valuable enantiomerically pure $β^2$- and $β^{2,2}$-amino ketones were obtained in satisfactory yields with good-to-excellent enantioselectivities by using chiral phosphoric acid catalysts to control the stereochemistry and chemoselectivity. Fluoro-heteroquaternary and full-carbon quaternary stereocenters that are challenging to prepare were successfully constructed.

[1] Key Laboratory of Natural Medicine and Immuno-Engineering of Henan Province, Henan University, Kaifeng, Henan 475004, China. [2] Singapore University of Technology and Design, Singapore 487372, Singapore. These authors contributed equally: Jiangtao Li, Manman Kong, Baokun Qiao. Correspondence and requests for materials should be addressed to Z.J. (email: chmjzy@henu.edu.cn)

The nucleophilic substitution of alkyl halides is a fundamental chemical transformation for precisely delivering molecular fragments to $sp^3$-hybridized carbon atoms using halide (X) as a directing group[1,2]. $S_N1$ and $S_N2$ reactions are two classic pathways, and they are ubiquitous in organic synthesis. However, the enantioselective version of the $S_N1$ approach for forming a stereogenic center at the carbon undergoing substitution by a simple alkyl halide, for which no special structural requirement for the removal of X is designated[3–5], remains as an unsolved problem due to the reaction mechanism. Meanwhile, the $S_N2$ reaction is stereospecific for which the generation of chiral products typically requires the utilization of the enantioenriched secondary alkyl halides. In this regard, Fu et al. made a major breakthrough by merging transition metal-catalyzed cross-coupling techniques with radical chemistry (Fig. 1a)[6]. In a bimetallic protocol such as this[7], achiral alkyl radicals are generated from both racemic alkyl halide reactants through inner-sphere single-electron transfer (SET) with a chiral transition metal catalyst (Mcat*). After transmetalation of the generated Mcat*X species with organometallic nucleophiles (NuM) and oxidative addition of the alkyl radical, the alkylmetal–Nu complex, which is produced as a single stereoisomer, undergoes reductive elimination to afford enantioconvergent substitution products. Thus far, this catalytic system has inspired the development of numerous chemical transformations with various NuM species[8–10]. Recently, Fu extended this protocol to nonmetallic secondary amine nucleophiles, where SET between a photosensitive metal–nucleophile complex and an alkyl halide

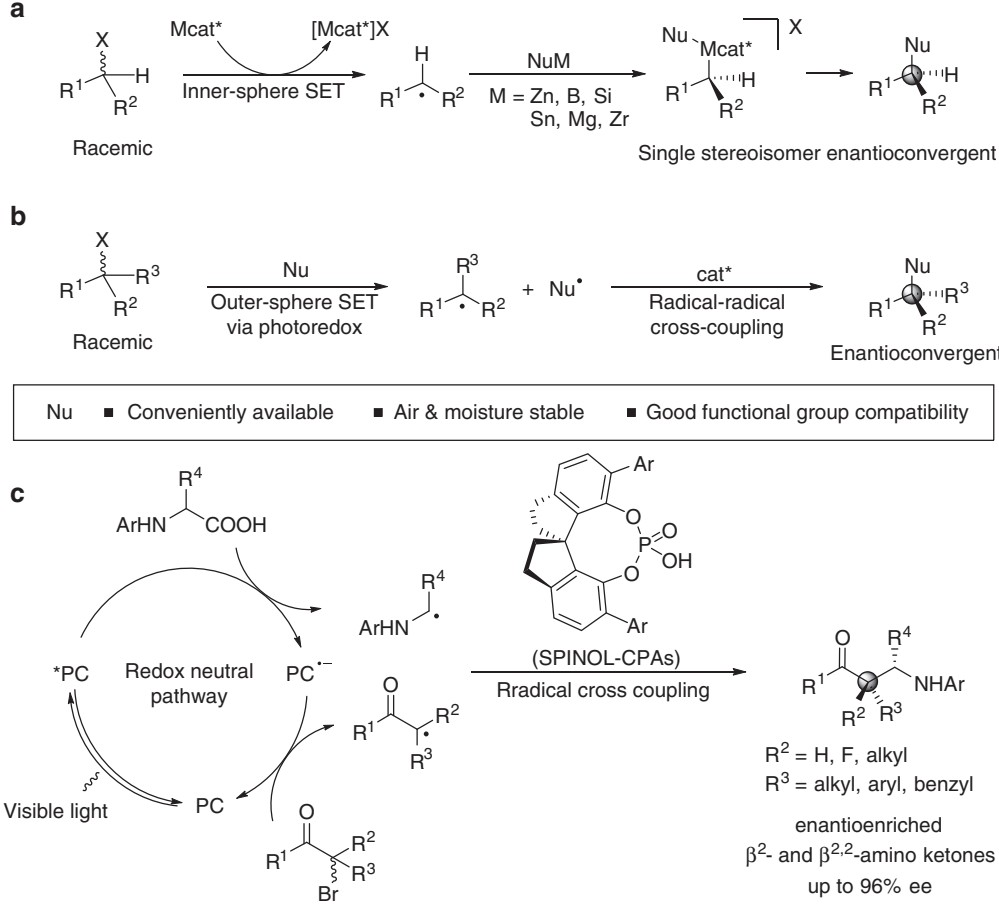

**Fig. 1** Strategies for catalytic asymmetric radical-based substitution reactions of simple alkyl halides. **a** Bimetallic protocol (known). **b** Radical coupling under visible light-induced photoredox catalysis for enantioconvergent substitution of alkyl halides (design profile). **c** A dual-catalysis approach to enantioselective radical coupling of N-aryl amino acids to α-bromoketones irradiated by visible light (this work). X halide. PC, photoredox catalyst. SPINOL-CPAs = 1,1′-spirobiindane-7,7′-diol (SPINOL)-based spirocyclic chiral phosphoric acids. With the recent rapid development of the field of visible light-mediated photoredox catalysis, this strategy has been seen as a powerful tool for achieving radical-based transformations[36,37]. This catalysis platform has found considerable success in outer-sphere SET oxidations of a wide array of readily available and bench-stable organic molecules as it provides diverse radical species and shows high functional group tolerance. Meanwhile, reductive dehalogenations of alkyl halides via electron transfer-fragmentation processes allows the generation of alkyl radicals[33,34]. In conjunction with our studies[19,20] on the development of enantioselective, dual-catalysis photoredox reactions[38–41], we speculated that radical coupling reactions[42,43] for connecting two distinct odd-electron partners using an extrinsic stereocontrol factor[44–46] would provide a complementary and structurally versatile approach for accessing the enantioconvergent products (**b**). This method would avoid the limitation of the narrow substrate scope faced by direct $S_H2$-based radical transformations[47], where stereoselectivity is also underdeveloped. Notably, the precise and absolute stereocontrol in a reaction with such a low-energy barrier and rapid bond-forming process is a formidable challenge, especially when building an uncommon full-carbon quaternary stereocenter. Here, we demonstrate a generic radical recombination strategy for the enantioselective aminoalkylation of α-bromoketones for modular $C_{sp3}$–$C_{sp3}$ bond formation (**c**). SPINOL-based spirocyclic chiral phosphoric acid (SPINOL-CPA) catalysts were used to control the chemoselectivity and stereoselectivity of the redox-neutral radical coupling between α-bromoketones and N-aryl amino acids. Both secondary α-bromoketones and tertiary (α-fluoro)-α-bromoketones were compatible with this α-aminoalkyl-substitution reaction, leading to valuable chiral $β^2$- and $β^{2,2}$-amino ketones with high enantioselectivities

**Table 1 Optimization of reaction conditions**

| Entry | Variation from the standard conditions | Yield (%)[a] | ee (%)[b] |
|---|---|---|---|
| 1 | None | 79 | 95 |
| 2 | C2 instead of C1 | 77 | 87 |
| 3 | C3 instead of C1 | 72 | 94 |
| 4 | C4 instead of C1 | 69 | 72 |
| 5 | C5 instead of C1 | 70 | 45 |
| 6 | C6 instead of C1 | 28 | 0 |
| 7 | No C1 | 30 | – |
| 8 | No DPZ | 20 | 94 |
| 9 | No light | 0 | – |

The reaction was performed on a 0.05 mmol scale. Entries 1–7, the chemical conversion of **1a** was >95% determined by crude [1]H NMR. Entry 8, the chemical conversion of **1a** was 23%
[a]Yields were determined from the isolated compound following chromatographic purification
[b]Enantiomeric excesses were determined by HPLC analysis on a chiral stationary phase

under visible light irradiation is responsible for the production of the alkyl radicals[11]. In this context, the capacity of the radical approach to C–X bond fragmentation has garnered substantial recognition for generating reactive species that can participate in asymmetric substitution reactions of simple alkyl halides.

## Results

**Reaction optimization**. The use of the non-fossil fuel-based α-amino acids as starting substrates always represents an attractive method in organic synthesis because of their environmentally benign features. In recent years, the groups of Das[12], Tan[13], MacMillan[14–16], and Rueping[17] successively reported the viability of generating the α-amino radicals from N-protected α-amino acids via facile visible light-driven single-electron oxidative decarboxylation. Inspired by these contributions, we began our study by exploring the model reaction between α-bromoketone **1a** and N-phenyl glycine (**2a**) with our developed metal-free[18] dicyanopyrazine-derived chromophore (DPZ)[19,20] as the photoredox catalyst (Table 1). Initial investigations using Stern–Volmer experiments revealed that the excited DPZ (*DPZ) species can facilitate single-electron oxidation of **1a**. Upon examining a range of reaction parameters, we observed that the reaction performed in 1,2-dimethoxyethane at 0 °C for 60 h in the presence of 0.5 mol % DPZ, 10 mol% chiral SPINOL-CPA[21] **C1**, and 3 Å molecular sieves affords a desired α-aminoalkyl-substitution product **3a** in 79% yield with 95% ee (enantiomeric excess) (entry 1). The substituents at the 6,6′-positions of SPINOL affected the enantioselectivity, and as an evidence, catalysts **C2–C4** showed different enantioselectivities for the formation of **3a** (entries 2–4). Catalyst BINOL-CPA[22–24] **C5** gave **3a** with 45% ee (entry 5), which confirmed the significant effect of the chiral backbone of

the catalyst on the enantioselectivity. The reaction was also examined using L-amino acid-based urea-tertiary amine bifunctional catalyst[20] **C6**, but the yield deteriorated and the reaction was not enantioselective (entry 6). In the absence of the chiral catalyst **C1**, **3a** was formed as an achiral mixture in only 30% of yield (>95% conversion of **1a**, entry 7). In both reactions (entries 6–7), the reductive debrominative protonation of **1a** became a primary transformation, given that ethyl phenyl ketone (**4**) was obtained as the major product. The reaction conducted in the absence of DPZ provided **3a** in only 20% of yield with 94% ee (23% conversion of **1a**, entry 8). These results suggested that the chiral catalyst is crucial for both the chemoselectivity and enantioselectivity, and the photocatalyst is necessary for the reactivity[25]. Control experiments confirmed that visible light is necessary for the transformation (entry 9).

**Substrate scope of secondary α-bromoketones**. To examine the scope of this radical-based enantioconvergent substitution reaction, optimal conditions were evaluated with a wide range of N-aryl amino acids and racemic secondary α-bromoketones for the synthesis of β²-amino ketones (Fig. 2). For the couplings of **1a** with N-aryl glycines **2a–f**, which feature electron-withdrawing or electron-donating substituents on the aryl ring, products **3a–f** were obtained in 69–80% yields with 94–95% ees. As for the scope of electrophiles, various secondary α-bromoketones with a variety of combinations of substituents (i.e., aryl, alkyl, and benzyl) at the 1- and 2-positions were compatible with the reaction conditions and provided the corresponding β²-amino ketones (**3g–w**) in 55–86% yields with 86–95% ees. Notably, the free hydroxyl group on the aromatic ring did not interfere with the reaction (product **3m**). To improve the enantioselectivity, catalyst **C2** was used to

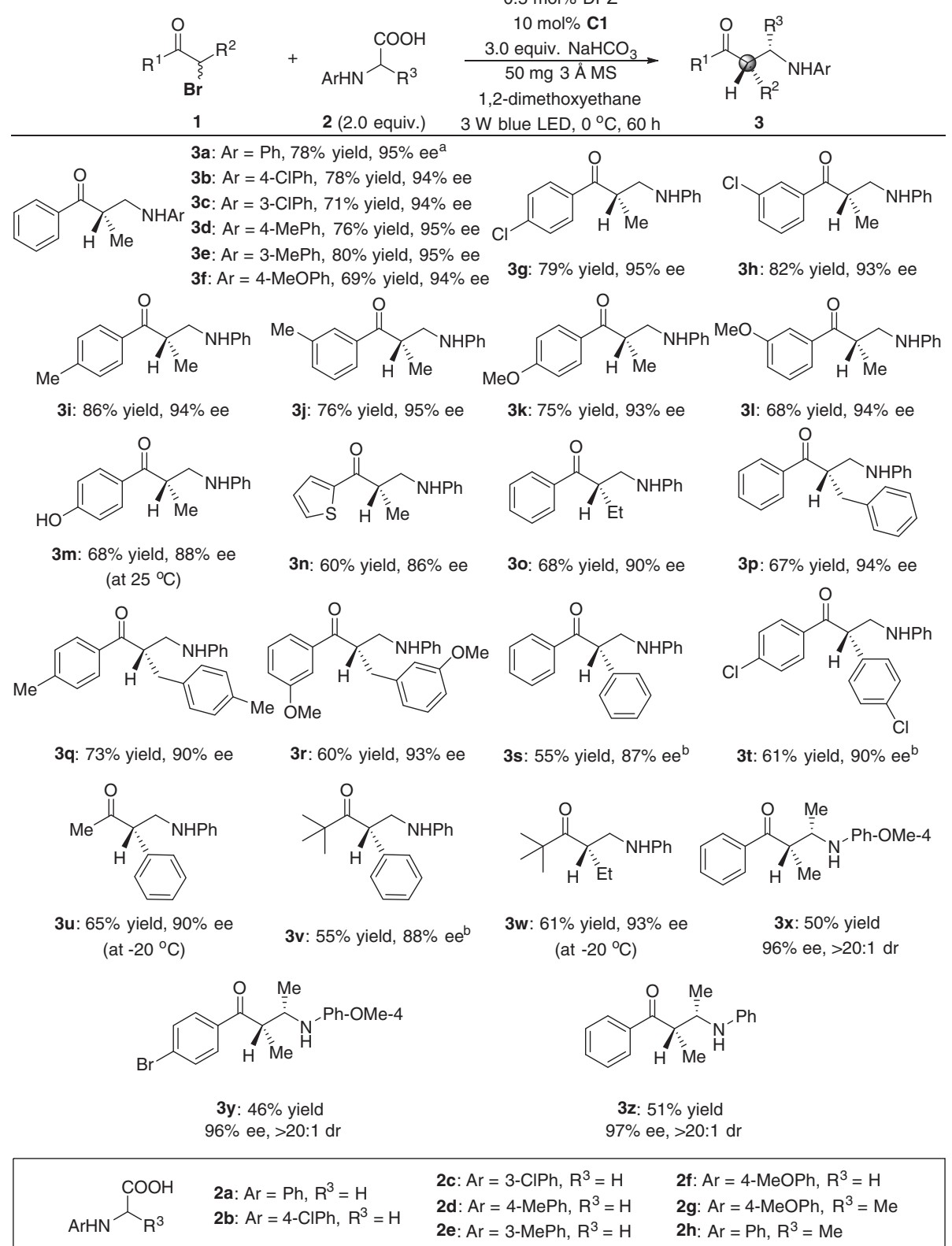

**Fig. 2** Substrate scope of *N*-aryl amino acids and secondary α-bromoketones. Reactions were performed with **1** (0.1 mmol), **2** (0.2 mmol), DPZ (0.5 × 10⁻³ mmol), **C1** (0.01 mmol), NaHCO₃ (0.3 mmol), and 3 Å MS (50 mg) in 1,2-dimethoxyethane (1.5 mL) at 0 °C. Yields were determined from the isolated material after chromatographic purification. Enantiomeric excesses were determined by HPLC analysis on a chiral stationary phase. [a]On a 1.0 mmol scale and 60 h, yield of **3a** = 75% and ee = 94% (2 × 3 W blue LED). [b]Catalyst **C2** was used instead of **C1**

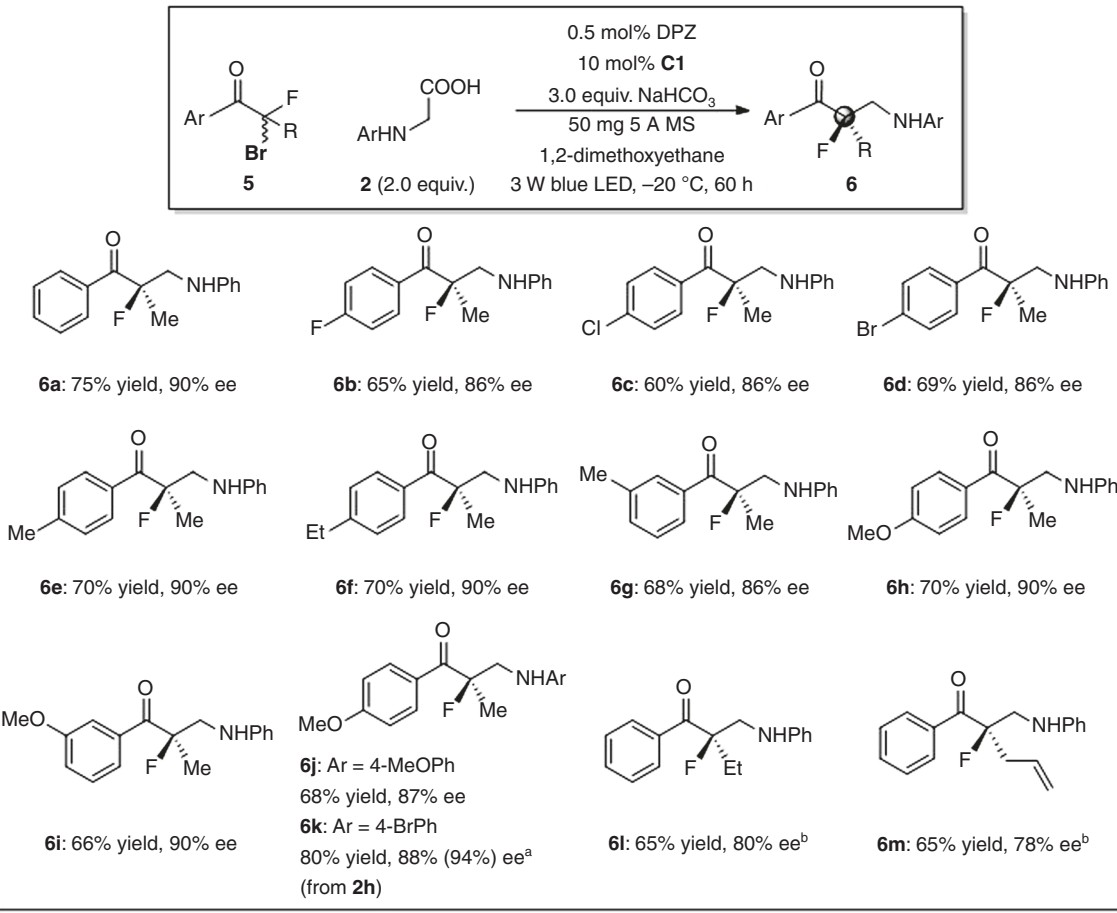

**Fig. 3** Substrate scope of tertiary α-bromo-α-fluoroketones with N-aryl glycines. Reactions were performed with **5** (0.1 mmol), **2** (0.2 mmol), DPZ (0.5 × 10⁻³ mmol), **C1** (0.01 mmol), NaHCO₃ (0.3 mmol), and 5 Å MS (50 mg) in 1,2-dimethoxyethane (1.5 mL) at −20 °C. Yields were determined from the isolated material after chromatographic purification. Enantiomeric excesses were determined by HPLC analysis on a chiral stationary phase. ᵃThe ee value in parenthesis was obtained after recrystallization. ᵇCatalyst **C3** (20 mol%) was used instead of **C1** and T = −45 °C

generate products **3s, t** and **3v**. The reaction to access **3a** was attempted on a 1.0 mmol scale, but under an irradiation by two 3 W blue LEDs, and the similar yield (75%) and enantioselectivity (94% ee) were observed. Reactions of **1** with N-MeOPh (**2g**) and N-Ph (**2h**) alanines were also carried out, and adducts **3x−z** containing two adjacent tertiary carbon stereocenters were obtained in 46–51% yields with 96–97% ees and >20:1 drs. A slightly decreased yield can be attributed to the decreased chemoselectivity of the coupling reactions. When examining other α-substituted N-aryl amino acids, this undesirable side reaction became dominant, and the reductive debrominative protonation product of **1** was observed almost exclusively.

**Substrate scope of α-bromo-α-fluoroketones.** Motivated by the potential applications of organofluorine compounds in a variety of scientific disciplines[26,27], the scope of α-bromo-α-fluoroketones **5** was examined next (Fig. 3). Under the established reaction conditions, but using 5 Å MS as an additive and at −20 °C, chiral β-fluoro-β-methyl amino ketones **6a–k** were prepared in 60–80% yields with 84–94% ees. When the α-substituent of **5** was ethyl and allyl, the enantioselectivities were lower. However, switching from catalyst **C1** to **C3** (20 mol%) and performing the reaction at a lower temperature (−45 °C) allowed products **6l–m** to be generated with good enantioselectivity.

**Substrate scope of tertiary α-bromoketones.** Encouraged by these results, we extended the generality of the reaction with

regard to racemic tertiary α-bromoketones **7** to address the challenge of forming full-carbon quaternary stereocenters (Fig. 4). Although the tertiary carbon radicals feature higher steric hindrance, the reactions proceeded smoothly with similar reactivity and chemoselectivity, and afforded β²,²-amino ketones **8a–i** in 60–70% yields. To achieve the best enantioselectivity, catalyst **C3** was used for the preparation of **8a–e** (82–92% ee), **C2** was used for **8f–h** (84–90% ee), and **C4** was used for **8i** (84% ee). Other chiral β²,²-amino ketones could be obtained with high enantioselectivity by tuning the 6,6′-substituents of the catalyst.

**Synthetic applications.** Enantiomerically pure β²- and β²,²-amino ketones are prevalent in biologically active compounds[28] and are direct precursors[29–32] to a number of important chiral structural motifs such as β²- and β²,²-amino acids, derivatives for peptide synthesis[30,31], and fluorohydrins[32]. For example, the treatment of β²,²-amino ketone **6j** with N,N′,N″-trichloroisocyanuric acid (**9**), followed by p-toluenesulfonyl chloride could replace the para-methoxyphenyl (PMB) N-protective group with a tosyl (Ts) group (Fig. 5). The resulting β²,²-amino ketone **10** was readily converted to fluorine-containing chiral ester **11** by a Baeyer–Villiger rearrangement with mCPBA. Meanwhile, the reduction of **6j** using TiCl₄ and BH₃•Me₂S in THF at −78 °C furnished fluorohydrin **12** with perfect diastereoselectivity and without erosion of the ee.

**Fig. 4** Substrate scope of tertiary α-bromoketones. Reactions were performed with **7** (0.1 mmol), **2** (0.2 mmol), DPZ (0.1 × 10$^{-3}$ mmol), **C2/C3/C4** (0.02 mmol), NaHCO$_3$ (0.3 mmol), and 5 Å MS (50 mg) in 1,2-dimethoxyethane (1.5 mL) at 0 °C. Yields were determined based on the isolated material after chromatographic purification. Enantiomeric excesses were determined by HPLC analysis on a chiral stationary phase

**Mechanistic studies**. Although the above results and previous reports[18,33,34] are consistent with a redox-neutral radical process triggered by the reductive quenching of DPZ\*, as depicted in Fig. 1c, control experiments were performed to further probe the reaction mechanism (Fig. 6). A radical-clock study using *cis*-cyclopropyl-substituted α-bromoketones **13** and **2a** was conducted first (Fig. 6a). β²-amino ketone **14** was obtained in 70% yield, and no ring-opened product was observed due to the maintained configuration of the cyclopropyl moiety. This result confirmed that the radical intermediates generated from the SET reductive debromination of α-bromoketones are very active and quickly couple with the α-amino radicals. The high reactivity of these alkyl radicals is consistent with the result of the reaction between **1a** and *N*-phenyl alanine (**2h**) in the presence of the radical scavenger TEMPO (1.5 equiv.), where product **15** was obtained in 55% yield via coupling with TEMPO (Fig. 6b). Notably, this reaction also generated products **4** and 4-amino tetrahydroquinoline **16** in 38% and 20% yields, respectively, but β²-amino ketone **3z** was not observed. Product **16** is likely generated from the Povarov reaction between the imine formed by two SET oxidations of **2i** and an enamine (the tautomer of the imine). Hence, the second reductive protonation of the alkyl radical must be fast and the substitution products should be formed by a radical cross-coupling reaction as the major operative process and not by a Mannich-type reaction of the α-ketone anion with an imine. Such an ionic pathway and a possible radical addition process could be further excluded by the control experiment shown in Fig. 6c, in which the reaction of α-bromoketone **7a** with *N*-phenyl glycine **2a** and *N*-MeOPh imine **17** derived from formaldehyde worked slightly sluggish, but very clean, only affording **8b** in 42% yield with 22% ee after 48 h and without **8a** being detected. Of note, the transformation of **7a** with *N*-MeOPh glycine **2f** under the reaction conditions furnished **8b** with 87% ee (Fig. 4). The different enantioselective results suggest

two distinct approaches to access **8b** in both reactions. Furthermore, while **8a** was not obtained, imine **18** through two single-electron oxidations of **2a** was not observed yet, which also eliminated the possibility of β²- and β²,²-amino ketones obtained from the transformations with imines in our reaction system. For higher concentration of product **8b** from imine **17**, it should be obtained from a Mannich-type radical addition than the α-amino radical generated from **2a**. In this process, the excellent reactivity that has been described by Knowles and co-workers[28] could be demonstrated by the result that no ketone **19** as the debrominative protonation product was detected. It is also worth mentioning that no reaction was detected when three kinds of the parent ketones with imine **17** were under the corresponding standard reaction conditions.

Chiral phosphoric acid catalysts were shown to provide higher selectivity for the radical coupling reaction over the reductive debrominative protonation (entries 1 and 7, Table 1). According to the persistent radical effect[35], it is plausible that phosphoric acid would act as a bifunctional H-bonding catalyst[29] to stabilize the electrophilic alkyl radical species and activate the nucleophilic α-amino radical variant, thus facilitating the selective C–C bond formation, but not the SET-redox reaction. The decreased yield with *N*-phenyl-*N*-methyl glycine as a starting material (45% yield, see Supplementary Table 3) and the $^{13}$C NMR analysis between substrate and catalyst (Supplementary Note 2) could be recognized as circumstantial evidences to demonstrate this hypothesized activation approach. This H-bonding effect offers another method of controlling the enantioselectivity of the formation of chiral substitution products. Additionally, a linear relationship was found between the ee of product **3t** and the ee of acid catalyst **C2** (Fig. 6d). This result suggests only a single molecule of chiral phosphate was involved in the crucial C–C bond-forming step. More investigations were continuously being conducted in our laboratory to make an accurate understanding of the mechanism.

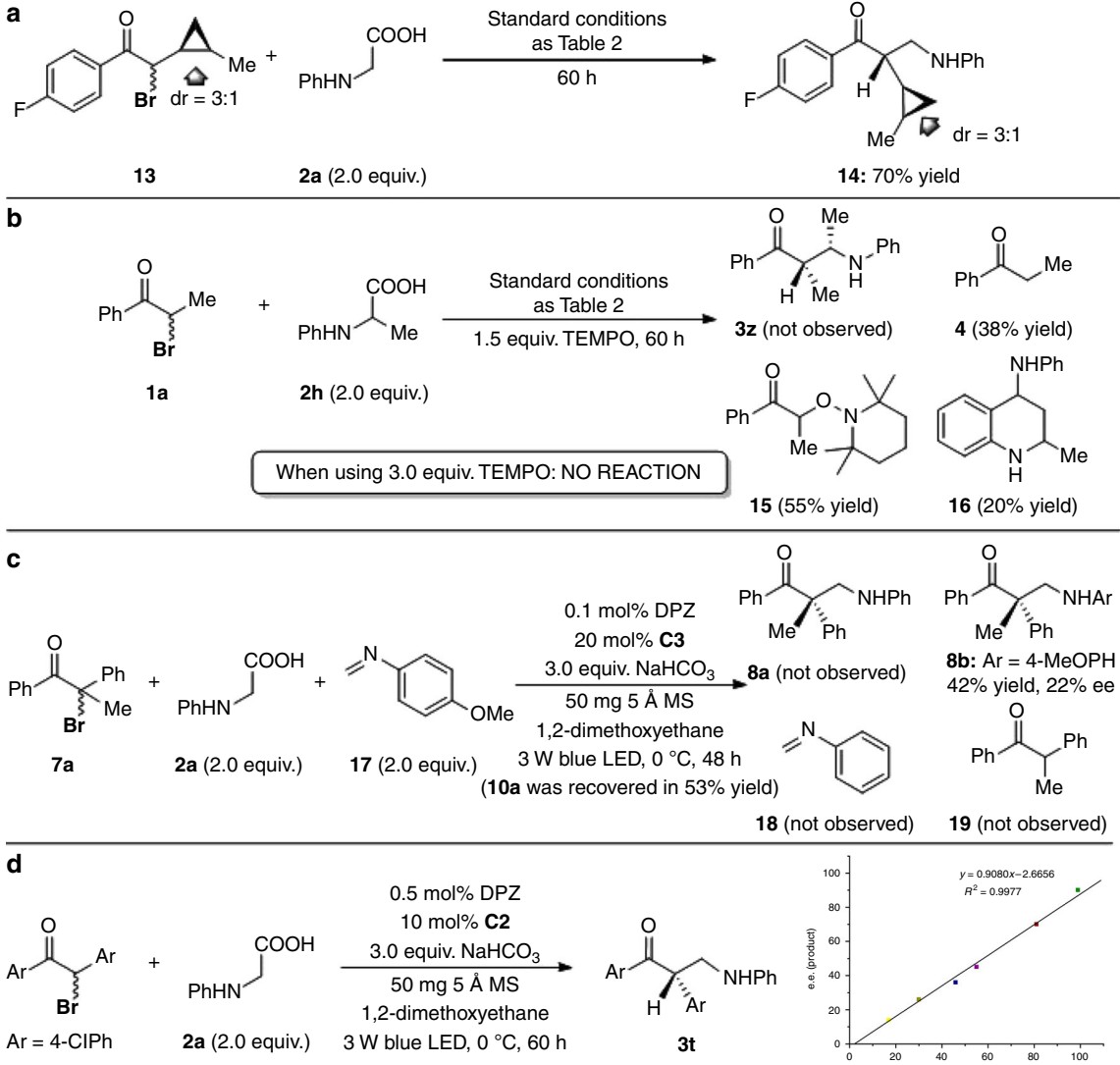

**Fig. 5** Synthetic utilities. The synthesis of β2-fluoro-β2-substituted amino ester **11** and fluorohydrin **12** from chiral β-fluoro-β-methyl amino ketone **6j**

**Fig. 6** Experimental studies to elucidate the mechanism. **a** Radical-clock examination. **b** Investigation on the effect of TEMPO. **c** The reaction of bromoketone **7a** with *N*-phenyl glycine **2a** and imine **17**. **d** The study on the relationship between ee of the CPA **C2** and ee of the product **3t**

## Discussion

In summary, we have developed a catalytic asymmetric photoredox radical-coupling method as a versatile strategy for the enantio-convergent substitution of alkyl halides. Alkyl radicals generated from the visible light-driven photoreductive dehalogenation of simple alkyl halides undergoing diverse radical-based transformations have been well-studied[33,34,36,37]. The high enantioselectivity in conjunction with broad substrate scope for Csp3–Csp3 bond

formation, especially the success in the construction of challenging fluoro-hetero-quaternary and full-carbon quaternary stereocenters, provides a crucial foundation for the pursuit of other valuable enantioconvergent substitution transformations of alkyl halides with readily available and bench-stable organic molecules.

## Methods

**General information**. For the NMR spectra of compounds in this manuscript, see Supplementary Figs. 1–80. For the HPLC spectra of compounds in this manuscript, see Supplementary Figs. 81–141. For details of optimization of reaction conditions, see Supplementary Tables 1–3. For the synthesis of substrates **13**, **17**, and **22**, see Supplementary Note 1. For details of mechanistic studies, see Supplementary Note 2. For details of synthetic applications, see Supplementary Note 3. For the determination of absolute configuration of products, sees Supplementary Note 4. For general information, general experimental procedure, and analytic data of compounds synthesized, see Supplementary Methods.

**Preparation of 3**. A total of 35.4 µL (0.0005 mmol, 0.005 equiv.) of DPZ solution (1.0 mg of DPZ in 200 µL of toluene) was added into a 10-mL Schlenk tube and then solvent was removed in vacuo. Subsequently, **1** (0.1 mmol, 1.0 equiv.), **2** (0.2 mmol, 2.0 equiv.), **C1** (0.01 mmol, 0.1 equiv.) for **3a–r, u, w, x, y** and **z** or **C2** (0.01 mmol, 0.1 equiv.) for **3s, t**, and **v**, 3 Å molecular sieves (50.0 mg), and NaHCO₃ (25.2 mg, 3.0 equiv.) in 1,2-dimethoxyethane (1.5 mL) were sequentially added and then degassed three times by the freeze-pump-thaw method. The reaction mixture was stirred under an argon atmosphere at 0 °C (the temperature was maintained in an incubator) for 30 min without light, and then irradiated by a 3 W blue LED (λ = 450–455 nm) from a 3.0 cm distance for another 60 h. The reaction mixture was directly loaded onto a short silica gel column, followed by gradient elution with petroleum ether/ethyl acetate (200/1~20/1 ratio). Removing the solvent in vacuo afforded products **3a–z**.

**Preparation of 6**. A total of 35.4 µL (0.0005 mmol, 0.005 equiv.) of DPZ solution (1.0 mg of DPZ in 200 µL of toluene) was added into a 10-mL Schlenk tube and then the solvent was removed in vacuo. Subsequently, **5** (0.1 mmol, 1.0 equiv.), **2** (0.2 mmol, 2.0 equiv.), **C1** (0.01 mmol, 0.1 equiv.) for **6a–k** or **C3** (0.02 mmol, 0.2 equiv.) for **6l** and **m**, 5 Å molecular sieves (50.0 mg), NaHCO₃ (25.2 mg, 3.0 equiv.) in 1,2-dimethoxyethane (1.5 mL) were sequentially added and then degassed three times by the freeze-pump-thaw method. The reaction mixture was stirred under an argon atmosphere at −20 °C for **6a–k** or at −45 °C for **6l** and **m** (the temperature was maintained in an incubator) for 30 min without light, and then irradiated by a 3 W blue LED (λ = 450–455 nm) from a 3.0 cm distance for another 60 h. The reaction mixture was directly loaded onto a short silica gel column, followed by gradient elution with petroleum ether/ethyl acetate (200/1~20/1 ratio). Removing the solvent in vacuo afforded products **6a–m**.

**Preparation of 8**. A total of 7.1 µL (0.0001 mmol, 0.001 equiv.) of DPZ solution (1.0 mg of DPZ in 200 µL of toluene) was added into a 10-mL Schlenk tube and then solvent was removed in vacuo. Subsequently, **7** (0.1 mmol, 1.0 equiv.), **2** (0.2 mmol, 2.0 equiv.), **C2** (0.02 mmol, 0.2 equiv.) for **8f–h**, **C3** (0.02 mmol, 0.2 equiv.) for **8a–e**, or **C4** (0.02 mmol, 0.2 equiv.) for **8i**, 5 Å molecular sieves (50.0 mg), NaHCO₃ (25.2 mg, 3.0 equiv.) in 1,2-dimethoxyethane (1.5 mL) were sequentially added and then degassed three times by the freeze-pump-thaw method. The reaction mixture was stirred under an argon atmosphere at 0 °C (the temperature was maintained in an incubator) for 30 min without light, and then irradiated by a 3 W blue LED (λ = 450–455 nm) from a 3.0 cm distance for another 60 h. The reaction mixture was directly loaded onto a short silica gel column, followed by gradient elution with petroleum ether/ethyl acetate (200/1~20/1 ratio). Removing the solvent in vacuo afforded products **8a–i**.

**Data availability**. The X-ray crystallographic coordinates for structures that support the findings of this study have been deposited at the Cambridge Crystallographic Data Centre (CCDC) with the accession code CCDC 1587114 (12), 1589705 (25), and 1814679 (26). The authors declare that all other data supporting the findings of this study are available within the article and Supplementary Information files, and also are available from the corresponding author upon reasonable request.

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

## Acknowledgements

Grants from NSFC (Nos. 21072044 and 21672052), Innovation Scientists and Technicians Troop Construction Projects of Henan Province, and Henan University are gratefully acknowledged. We also appreciate Miss Xinyi Ye and Prof. Choon-Hong Tan (NTU) for their generous help in the analysis of HRMS data and Mr. Yangyang Shen (ICIQ) for constructive discussions.

## Author contributions

Z.J. conceived and designed the experiments. J.L., M.K., and B.Q. performed the experiments and prepared the Supplementary Information. R.L. and X.Z. helped with isolating the new compounds and analyzing the data. Z.J. wrote the paper. J.L., M.K., and B.Q. contributed equally to this work. All authors discussed the results and commented on the manuscript.

## Additional information

**Competing interests:** The authors declare no competing interests.

