## [Peer Review File · Nature Communications]

Reviewer #1 (Remarks to the Author):

The manuscript reports a dual catalytic system, based on the combined action of a chiral phosphoric acid organocatalyst and a visible-light activated dicyanopyrazine-derived photocatalyst (DPZ), that can control the radical coupling of transiently generated open-shell species. This redox-neutral mechanism is based on the ability of the photoredox catalyst to generate two transient C-sp³ radicals, whose coupling is under the stereocontrol of the chiral phosphoric acid. The net reaction leads to the stereoconvergent formation of chiral molecules from readily available racemic alpha-bromo ketones and N-aryl aminoacids. In general, this study provides a solution to the difficult target of achieving precise stereocontrol of radical coupling processes, which is greatly complicated by its intrinsic high rate. Besides this conceptual implication, the scope of the reaction is good and the produced chiral beta-amino ketones are synthetically valuable. In addition, the method is suitable for the formation of valuable carbon quaternary stereocenters and fluorine-containing stereocenters. On this basis, the manuscript presents interesting results that can attract the interest of the wide readership of the journal. However, it contains many scientific inaccuracies that need to be adequately amended. In addition, some mechanistic aspects have not been adequately addressed.

The paper is disseminated of scientific inaccuracies and mistakes. First and foremost, all along the manuscript, including the title, the authors describe their reactivity as a "radical nucleophilic substitution reaction", a definition which is an inherent contradiction. It is worth to remember that nucleophilic substitutions are heterolytic reactions, in which the cleavage of the covalent bond leads to two charged intermediates. In this case the mechanism, as proposed by the authors, involve a radical-radical coupling, with no involvement of charged intermediates. It is highly inaccurate to create not-existing parallelisms between radical and polar reactions. Indeed radicals are neutral open-shell intermediates and their reactivity is completely different when compared to polar intermediates. Radicals can undergo Homolytic Substitution (SH₂) reactions, not nucleophilic substitution. In addition, the reported chemistry likely proceeds via a radical coupling mechanism, thus also the SH₂ manifold does not apply here.

Sentences like the following (Title and abstract) ..." we report [...] a strategy for enantioconvergent radical nucleophilic substitution reactions"... or (page 3, last sentence) ..."radicals that can participate in asymmetric nucleophilic substitution reactions..." or (page 5) ..."this aminoalkyl-nucleophilic substitution reaction" are alarming, to say the least, and should be changed.

Another alarming point concerns the way the SN₂ reaction is discussed. The very first sentence of the abstract reads: "Classic nucleophilic substitution reactions of alkyl halides (SN₁ and SN₂) are not generally amenable to the enantioselective construction of a stereogenic center at the carbon undergoing substitution". By definition, the SN₂ reaction proceeds in a very predictable manner, since it is stereospecific: substitution occurs with inversion of stereochemistry, resulting from the 'backside attack' of the electrophilic carbon by the nucleophile. This is one of the earliest pieces of mechanistic organic chemistry taught to undergraduates worldwide.

This means that SN₂ processes are intimately linked to stereocontrol (they are stereospecific). Probably the authors mean that it is difficult to design a stereoconvergent catalytic reaction starting from racemic alkyl halides when the SN₂ manifold is in operation. The way it reads now, the introductory section is delivering an erroneous message and should be changed.

Table 1 - numbers in parenthesis indicate conversion, as estimated by NMR. Given the detection limit of NMR spectroscopy (about 95% signal/error sensitivity), writing 100% is not pertinent. In addition, the stoichiometry of the reaction is not specified, which makes it difficult to understand which is the limiting reagent and to what the conversion refers.

The reactions have been conducted on a very small scale (0.1 mmol). The authors should demonstrate that a synthetically meaningful scale is possible, performing the process at least at 1 mmol scale.

The following paper (Chem. Eur. J. 2016, 22, 13464), where the amino-radicals are generated from the same substrate used in this study and using a photoredox catalyst, should be adequately cited and discussed in the text. The following study, reporting on a photochemical protocol to set carbon-fluorine stereogenicity with high fidelity, has been overlooked: ACIE 2017, 56, 11875.

About the mechanism: the real role of the chiral phosphoric catalyst is intriguing. The results in entry 6, table 1 indicates that the acid catalyst is responsible of channeling the process toward the radical coupling pattern, avoiding H abstraction from the generated alpha-keto radical. On this basis, the proposal advanced by the authors, that the chiral acid stabilises the alpha-carbonyl radical by H-bonding interaction, is reasonable. Albeit further demonstrations to support the persistency of this intermediate would be welcome, at the present stage of investigation the proposed mechanism is sounding.

Finally, the authors used compound 13 as radical-clock and, after observation that the closed compound 14 is the only product of the reaction, concluded that "radical intermediates generated from the SET reductive debromination of α -bromoketones are very active and quickly couple with the α -amino radicals". It is however known that similar unsubstituted cyclopropane undergoes a rapid ring-closing, which is faster than the ring-opening event. For example, in the cyclopropylbenzyl radical the ring opening process, albeit reasonably fast, is reversible with the cyclic product being favored by two orders of magnitude (see: J. Chem. Soc., Chem. Commun. 1990, 923-925 and J. Am. Chem. Soc. 2000, 122, 2988-2994). Considering the higher stability of the originally formed secondary radical with respect to the primary radical, which originates upon ring-opening, along with the above-mentioned equilibrium, it is no surprise that only product 14 could be detected (trapping of the more stable secondary radical).

To sum up, this is a potentially interesting manuscript. However, many points have to be carefully addressed in order to raise the level of accuracy and reach the standards expected from a high-profile scientific publication.

Reviewer #2 (Remarks to the Author):

Jiang and co-workers reported an exciting radical strategy for asymmetric nucleophilic substitution through visible-light induced photoredox catalysis. The combination of a photoredox catalyst i.e. DPZ and a chiral SPINOL-CPA efficiently promoted the transformations between a variety of α -bromoketones and N-aryl α -amino acids. Besides the excellent enantioselectivities obtained, the elusive construction of all-carbon quaternary and fluoro-hetero-quaternary stereocenters has been successfully realized. The substrate scope is broad definitely. There are several important breakthroughs in this work: 1) photoredox catalysis has been demonstrated as an efficient strategy to enantioselective nucleophilic substitution for the first time. The generation of radical species from readily available and bench-stable organic nucleophiles through this catalytic platform has been well-established. Thus, the referee believes that this protocol should open a new avenue for the pursuit of other useful radical nucleophilic transformations with diverse competent nucleophiles. Given that the radical species generated from alkyl halides has been extensively used in photoredox catalytic transformations, the development of enantioselective patterns becomes promising; 2) the first α -aminoalkyl-substitution of alkyl halides has been developed in asymmetric nucleophilic substitution; 3) the construction of an all-carbon quaternary or fluoro-hetero-quaternary stereocenter has been firstly accomplished in the areas of asymmetric nucleophilic substitution and the radical-radical cross-coupling via asymmetric photoredox catalysis; 4) this is an unprecedented example of building chiral fluoro-hetero-quaternary stereocenters with excellent enantioselectivities in asymmetric photoredox catalysis; 5) chiral β 2- and β 2,2-amino ketones are very important molecular architectures and the catalytic synthetic method is still lack. Undoubtedly, the obtained valuable products further robustly increase the quality of this work. The

reasonable and ingenious transformations of fluorinated products 6 to fluoro-containing β 2,2-amino esters and fluorohydrins verify the important value of this methodology well; 6) asymmetric photoredox catalysis enabled by chiral Lewis acids has been well established, but the transition-metal-free catalytic system including photoredox catalyst and chiral catalyst is rare and significant for the sustainable properties. I agree the authors' suggestion that this work would open a new paradigm for asymmetric organocatalysis. In this context, the referee strongly supports this significant progress in both asymmetric nucleophilic substitution reaction and asymmetric photoredox catalysis to be published in Nature Communications. Few corrections are required to address:

1)Page 5, "SH2-based" should be "SN2-based".

2)Page 7, according to the description clue, "the reaction could also be conducted..." should be "the reaction was also examined".

3)One relative literature on the asymmetric radical cross coupling in photoredox catalysis is missed, please add: J. Am. Chem. Soc. 2017, 139, 17245–17248 by Eric Meggers.

Reviewer #3 (Remarks to the Author):

In this manuscript, Jiang and coworkers demonstrate the catalytic, enantioselective coupling of two radicals in an intermolecular C-C coupling. The electrophilic radical partner is derived from alpha-halo-carbonyls, while the nucleophilic radical precursor is an amino acid that is decarboxylated. Eventually the two radicals (alpha-carbonyl and alpha-amino) combine in a redox-neutral fashion to afford a range of Mannich reaction-type products with excellent yields and selectivities. In certain cases, alpha branched amino acids are employed to form products with two stereocenters in high diastereoselectivity. On the alpha-halocarbonyl side, alpha-fluoro-radicals, as well as tertiary-radicals to make quaternary centers, are both also amenable to this protocol. In a pair of synthetic applications, the products are converted to beta-amino-acids, or 1,3-amino-alcohols - each with high stereoselectivity. Mechanistic experiments are included to offer further insight. The descriptions of the reaction optimization and characterization data included in the SI make this paper of high scientific quality.

Major issues/concerns:

1 When employing a cyclopropyl radical clock (Fig 5a), no ring-opened product is observed. The authors suggest this is because "SET reductive debromination of α -bromoketones are very active and quickly couple with the α -amino radicals." This is an unlikely explanation. In fact, Fig 5b disproves it. A better radical clock is the cis-phenyl-substituted cyclopropane, which can reversibly open-and-close because of the benzyl stabilization of the open-formed benzyl radical, but this opening can be detected by measuring the final d.r. of the cyclopropane ring within the product. For an example, see: J. Am. Chem. Soc. 2010, 132, 10012

2. Since the products of this paper are Mannich reaction products, it is imperative to rule out a 2-electron (Mannich reaction) mechanism. However, the experiment in Fig 5c is not good enough since the PMP-imine is not the same one used throughout the paper. 17 should be replaced with the same Ph-formaldehyde-imine that could otherwise be the intermediate throughout.

3. There is no mention of how the catalyst is controlling selectivity. Since this is a major aspect of the paper, some experiments should be provided. NMR binding studies with each component (halide vs amino acid) would at least suggest which component is likely to bind to the catalyst.

Reviewer 1' comments and our responses

- 1) The paper is disseminated of scientific inaccuracies and mistakes. First and foremost, all along the manuscript, including the title, the authors describe their reactivity as a “radical nucleophilic substitution reaction” , a definition which is an inherent contradiction. It is worth to remember that nucleophilic substitutions are heterolytic reactions, in which the cleavage of the covalent bond leads to two charged intermediates. In this case the mechanism, as proposed by the authors, involve a radical-radical coupling, with no involvement of charged intermediates. It is highly inaccurate to create not-existing parallelisms between radical and polar reactions. Indeed radicals are neutral open-shell intermediates and their reactivity is completely different when compared to polar intermediates. Radicals can undergo Homolytic Substitution (SH2) reactions, not nucleophilic substitution. In addition, the reported chemistry likely proceeds via a radical coupling mechanism, thus also the SH2 manifold does not apply here. Sentences like the following (Title and abstract) “...” we report [...] a strategy for enantioconvergent radical nucleophilic substitution reactions” ... or (page 3, last sentence) “...” radicals that can participate in asymmetric nucleophilic substitution reactions...” or (page 5) “...” this aminoalkyl-nucleophilic substitution reaction” are alarming, to say the least, and should be changed.

Response: The authors agree and appreciate this very useful guidance. After careful considerations, the title has been revised as ‘Enantioconvergent Substitution of Alkyl Halides via Catalytic Asymmetric Photoredox Radical-Radical Cross-Coupling’. The corresponding contents as mentioned by the referee have also been revised accordingly.

- 2) Another alarming point concerns the way the S_N2 reaction is discussed. The very first sentence of the abstract reads: “Classic nucleophilic substitution reactions of alkyl halides (S_N1 and S_N2) are not generally amenable to the enantioselective construction of a stereogenic center at the carbon undergoing substitution” . By definition, the S_N2 reaction proceeds in a very predictable manner, since it is stereospecific: substitution occurs with inversion of stereochemistry, resulting from the ‘backside attack’ of the electrophilic carbon by the nucleophile. This is one of the earliest pieces of mechanistic organic chemistry taught to undergraduates worldwide. This means that S_N2 processes are intimately linked to stereocontrol (they are stereospecific). Probably the authors mean that it is difficult to design a stereoconvergent catalytic reaction starting from racemic alkyl halides when the S_N2 manifold is in operation. The way it reads now, the introductory section is delivering an erroneous message and should be changed.

Response: In the abstract, the corresponding sentence (the first paragraph) has been revised as ‘Classic nucleophilic substitution reactions (S_N1 and S_N2) are not generally amenable to the enantioselective variants that use simple and **racemic** alkyl halide electrophiles.’ Furthermore, according to the referee’s suggestions, the introduction has been revised, in which S_N2 reaction was described separately as: ‘Meanwhile, the S_N2 reaction is stereospecific for which the generation of chiral products typically requires the

utilization of the enantioenriched secondary alkyl halides.’ (See page 3 in the revised manuscript). We sincerely thank the referee for this pertinent advice.

- 3) Table 1 - numbers in parenthesis indicate conversion, as estimated by NMR. Given the detection limit of NMR spectroscopy (about 95% signal/error sensitivity), writing 100% is not pertinent. In addition, the stoichiometry of the reaction is not specified, which makes it difficult to understand which is the limiting reagent and to what the conversion refers.

Response: The conversion of 100% has been revised as >95%. To make it easier to understand, the results were summarized as the footnote of Table 1, in which the chemical conversion was specified with the starting substrate **1a** as the limiting reagent (See page 6 in the revised manuscript).

- 4) The reactions have been conducted on a very small scale (0.1 mmol). The authors should demonstrate that a synthetically meaningful scale is possible, performing the process at least at 1 mmol scale.

Response: We have attempted the reaction to access **3a** on a 1.0 mmol. The reaction worked smoothly when irradiated by two 3W blue LEDs, affording **3a** in 75% yield with 94% ee. The results demonstrate the synthetic value of this work. The results were added in Fig. 2 and the descriptions were presented accordingly (See page 8 and 9 in the revised manuscript).

- 5) The following paper (Chem. Eur. J. 2016, 22, 13464), where the amino-radicals are generated from the same substrate used in this study and using a photoredox catalyst, should be adequately cited and discussed in the text. The following study, reporting on a photochemical protocol to set carbon-fluorine stereogenicity with high fidelity, has been overlooked: ACIE 2017, 56, 11875.

Response: The CEJ paper has been cited as the reference 37 in the revised manuscript and an article on chiral C-F bond formation (ACIE 2017, 56, 11875) has also been added (designated as ref. 44 in the revised manuscript).

- 6) The authors used compound 13 as radical-clock and, after observation that the closed compound 14 is the only product of the reaction, concluded that “radical intermediates generated from the SET reductive debromination of α -bromoketones are very active and quickly couple with the α -amino radicals”. It is however known that similar unsubstituted cyclopropane undergoes a rapid ring-closing, which is faster than the ring-opening event. For example, in the cyclopropylbenzyl radical the ring opening process, albeit reasonably fast, is reversible with the cyclic product being favored by two orders of magnitude (see: J. Chem. Soc., Chem. Commun. 1990, 923-925 and J. Am. Chem. Soc. 2000, 122, 2988–2994). Considering the higher stability of the originally formed secondary radical with respect to the primary radical, which originates upon ring-opening, along with the

above-mentioned equilibrium, it is no surprise that only product **14** could be detected (trapping of the more stable secondary radical).

Response: The *cis*-cyclopropyl-substituted α -bromoketone **13** has been synthesized accordingly. The reaction of **13** and **2a** was performed under the standard reaction conditions. It was found that the configuration of such cyclopropyl moiety in the obtained product **14** was maintained (See page 13 and 14 in the revised manuscript. The corresponding experiments could be checked in Supplementary Note 2 [page S109-S110 in the revised SI] and Supplementary Note 2 [page S111-S112 in the revised SI]). The authors sincerely appreciate the referee for this helpful suggestion.

Reviewer 2' comments and our responses

- 1) Page 5, "SH2-based" should be "SN2-based".

Response: This is SH2 but not SN2. No revision is suggested.

- 2) Page 7, according to the description clue, "the reaction could also be conducted..." should be "the reaction was also examined".

Response: Done (See page 7 in the revised manuscript). Thanks!

- 3) One relative literature on the asymmetric radical cross coupling in photoredox catalysis is missed, please add: J. Am. Chem. Soc. 2017, 139, 17245–17248 by Eric Meggers.

Response: This paper has been added (designated as ref. 25 in the revised manuscript)

Reviewer 3' comments and our responses

- 1) When employing a cyclopropyl radical clock (Fig 5a), no ring-opened product is observed. The authors suggest this is because "SET reductive debromination of α -bromoketones are very active and quickly couple with the α -amino radicals." This is an unlikely explanation. In fact, Fig 5b disproves it. A better radical clock is the *cis*-phenyl-substituted cyclopropane, which can reversibly open-and-close because of the benzyl stabilization of the open-formed benzyl radical, but this opening can be detected by measuring the final d.r. of the cyclopropane ring within the product. For an example, see: J. Am. Chem. Soc. 2010, 132, 10012.

Response: This comment is same as the referee 1's. The *cis*-cyclopropyl-substituted α -bromoketone **13** has been synthesized accordingly. The reaction of **13** and **2a** was performed under the standard reaction conditions. It was found that the configuration of such cyclopropyl moiety in the obtained product **14** was maintained (See page 13 and 14 in the revised manuscript. The corresponding experiments could be checked in Supplementary Note 2 [page S109-S110 in the revised SI] and Supplementary Note 2 [page S111-S112 in the revised SI]). We also appreciate this referee for the instructive suggestion.

- 2) Since the products of this paper are Mannich reaction products, it is imperative to rule out a 2-electron (Mannich reaction) mechanism. However, the experiment in Fig 5c is not good enough since the PMP-imine is not the same one used throughout the paper. **17** should be replaced with the same Ph-formaldehyde-imine that could otherwise be the intermediate throughout.

Response: In Fig. 2, the authors have exhibited the use of *N*-MeOPh alanine (**2g**) as the starting substrate, and the corresponding substitution product **3x** was obtained in 50% yield with 96% ee and >20:1 dr (See page 8 in the revised manuscript). When **2g** was oxidized twice via a reductive debromination process, the PMP-imine **17** should be generated. In this context, the authors believe that this experiment can satisfy the request of the referee.

- 3) There is no mention of how the catalyst is controlling selectivity. Since this is a major aspect of the paper, some experiments should be provided. NMR binding studies with each component (halide vs amino acid) would at least suggest which component is likely to bind to the catalyst.

Response: In the reaction system, α -bromoketone generates an alkyl radical species and *N*-aryl amino acid presents an α -amino radical. Accordingly, we carried out the NMR binding studies using acetophenone and *N*-methylaniline as the substrate model. It was found that the ¹³C signal peaks of ketone of acetophenone and methyl of *N*-methylaniline were shifted to the downfield (for C=O of acetophenone, from 198.1409 to 198.2307 ppm; for methyl of *N*-methylaniline, from 26.5850 to 37.4158 ppm). The results indicate the potentially viability of the H-bonding interactions of CPA with the two substrates. As such, a bifunctional catalysis of CPA is plausible. The corresponding descriptions have been revised as ‘According to the persistent radical effect,⁴⁵ it is plausible that the phosphoric acid would act as a bifunctional H-bonding catalyst to stabilize the electrophilic alkyl radical species and activate the nucleophilic α -amino radical variant, thus facilitating the selective C–C bond formation but not the SET redox reaction. The decreased yield with *N*-phenyl-*N*-methyl glycine as the starting material (45% yield, see Supplementary Table S3) and the NMR binding studies (See Supplementary Note 2) could be recognized as circumstantial evidences to demonstrate this hypothesized activation approach.’. More investigations were continuous conducting in our laboratory to make an accurate understanding of the mechanism.

Reviewer #1 (Remarks to the Author):

In the revised manuscript, the authors have adequately addressed many of the concerns raised during the previous round of evaluation. Now the paper is reaching the level of accuracy required for publication.

Still, there are few aspects to further consider and address:

- The new title still presents some inaccuracies: the authors continue to mention a substitution path, which is not pertinent in radical chemistry. This nomenclature should be avoided. In addition, this chemistry is not a "cross-coupling" reaction, but a radical coupling process. Cross coupling is generally reserved to metal-mediated bond-forming processes.

- Finally, the following pertinent manuscript, which recently appeared on line, should be cited: Science, 2018, eaar6376.

Reviewer #3 (Remarks to the Author):

Despite the author's correcting of the minor issues, the most important scientific issue remains unsolved:

Figure 5c is still an inconclusive experiment that does not preclude an ionic pathway. Since this is the most crucial mechanistic experiment of the paper (to demonstrate the validity of the proposed, unique radical-coupling pathway), more needs to be done. For example, it seems likely that if the Et₃N is removed (leaving the Bronsted acid unquenched), either 1a (or the reduced Ph-Et ketone) would participate in the Mannich with the imine (also, the authors' explanation for using the wrong imine does not make much sense, and suggests some negative data is not being shared). Ultimately, the likely ionic (i.e. Mannich reaction) mechanistic pathway is simply too precedented to ignore (especially via this type of Bronsted activation of such imines). Better controls are needed to ensure a two-electron pathway is not in fact the major operative mechanism.

All other revisions are satisfactory.

Dr. Zhiyong Jiang
Key Laboratory of Natural Medicine and Immuno-Engineering of Henan Province
Henan University, Jinming Campus
Kaifeng, Henan, P. R. China, 475004
Tel./Fax: +86-371-2286-4665
E-mail: chmjzy@henu.edu.cn

Dear Dr. Giovanni Bottari,

Re: Formal Enantioconvergent Substitution of Alkyl Halides via Catalytic Asymmetric Photoredox Radical-Radical Cross-Coupling by *Jiangtao Li, Manman Kong, Baokun Qiao, Richmond Lee, Xiaowei Zhao, and Zhiyong Jiang**

First of all, I appreciate the constructive comments from the referees. I agree all of them and the manuscript has been revised as outlined below. According to your instructions, the revised contents are highlighted in the manuscript by giving the text a yellow background.

Point to Point Response to Reviewers' Comments

Please take note that all the descriptive, positive comments of the reviews are omitted, and only reviewer's comments expressing their concerns/suggestions are listed below, which are followed by our response.

Reviewer 1' comments and our responses

- 1) The new title still presents some inaccuracies: the authors continue to mention a substitution path, which is not pertinent in radical chemistry. This nomenclature should be avoided. In addition, this chemistry is not a "cross-coupling" reaction, but a radical coupling process. Cross coupling is generally reserved to metal-mediated bond-forming processes.

Response:

For the 'substitution', it is the reaction result as the Br of alkyl bromides was substituted by α -aminoalkyls. Actually, in the nomenclature of SH2 reaction, its full name is Bimolecular Homolytic *Substitution* which is a radical process but not a heterolytic reaction. In this context, we think this description seems suitable. To make the title more reasonable as the referee required, a 'Formal' was added.

For 'radical-radical cross-coupling', this is one of the radical coupling reactions in the radical chemistry and another one is radical-radical homo-coupling. Many literatures have used this description in radical reactions, please check: Xie, J., Jin, H. & Hashimi, A. S. K. The Recent Achievements of Redox-Neutral Radical C–C Cross-Coupling Enabled by Visible-Light. *Chem. Soc. Rev.* **46**, 5193–5203 (2017). (ref. 22); Wang, C. et al. Asymmetric Radical–Radical Cross-Coupling through Visible-Light-Activated Iridium Catalysis. *Angew. Chem. Int. Ed.* **55**, 685–688 (2016). (ref. 24); Studer, A. The Persistent

Radical Effect in Organic Synthesis. *Chem. Eur. J.* **7**, 1159–1164 (2001). (ref. 46) and others, such as: Huang, Z. et al. Radical-Radical Cross-Coupling for C-S Bond Formation. *Org. Lett.* **18**, 2351–2354 (2016); Amador, A. G. & Yoon, T. P. A Chiral Metal Photocatalyst Architecture for Highly Enantioselective Photoreactions. *Angew. Chem. Int. Ed.* **55**, 2304–2306 (2016). And so on.

- 2) The following pertinent manuscript, which recently appeared on line, should be cited: *Science*, 2018, eaar6376.

Response: This very nice *Science* paper has been cited as the reference 43 in the revised manuscript.

Reviewer 3' comments and our responses

- 1) Figure 5c is still an inconclusive experiment that does not preclude an ionic pathway. Since this is the most crucial mechanistic experiment of the paper (to demonstrate the validity of the proposed, unique radical-coupling pathway), more needs to be done. For example, it seems likely that if the Et₃N is removed (leaving the Bronsted acid unquenched), either **1a** (or the reduced Ph-Et ketone) would participate in the Mannich with the imine (also, the authors' explanation for using the wrong imine does not make much sense, and suggests some negative data is not being shared). Ultimately, the likely ionic (i.e. Mannich reaction) mechanistic pathway is simply too precedented to ignore (especially via this type of Bronsted activation of such imines). Better controls are needed to ensure a two-electron pathway is not in fact the major operative mechanism.

Response: Yes, the authors agree the referee that the experiments shown in Figure 5c are not enough to preclude an ionic pathway, which has been deleted accordingly. Indeed, the results displayed in Figure 5b are the better control to robustly demonstrate the radical-radical cross-coupling as the major operative process:

The use of 1.5 equiv. TEMPO could inhibit the generation of the desired product **3z**, and reductive dehalogenative protonation product **4** and 4-amino tetrahydroquinoline **16** were obtained in 38% and 20% yields. The formations of **4** and **16** indicate the existence of an α -ketone anion and an imine derived from **2i**. If the ionic Mannich reaction plays a major role, the product **3z** should be detected since the phosphoric acid catalyst exists in the

reaction system. Therefore, the reaction between α -ketones and imines does not have so high reactivity under the established reaction conditions. Actually, the cross-coupling of the similar distinct two radical species has been demonstrated by Cheng and co-workers. Please find the reference: Chen, W. et al. Building Congested Ketone: Substituted Hantzsch Ester and Nitrile as Alkylation Reagents in Photoredox Catalysis. *J. Am. Chem. Soc.* **138**, 12312–12315 (2016). (ref. 23). Hence, the second reductive protonation of the alkyl radical must be fast, and the substitution products should be formed by a radical cross-coupling reaction as the major operative process and not by a Mannich-type reaction of the α -ketone radical or anion with an imine.

Both the above-mentioned possible pathways could be further excluded by the control experiment shown in Fig. 5C, in which the reaction of α -bromoketone **10a** with *N*-phenylglycine **2a** and *N*-MeOPh imine **22** derived from formaldehyde worked slightly sluggish but very clean, only affording **12b** in 42% yield with 22% ee after 48 hours and without **12a** detected. Of note, the transformation of **10a** with *N*-MeOPh glycine **2f** furnished **12b** with 87% ee under the reaction conditions (See Fig. 4). The different enantioselective results suggest two distinct approaches to access **12b** in both reactions. Furthermore, while **12a** was not obtained, imine **23** through twice single-electron oxidations of **2a** was not observed yet, which also could eliminate the possibility of β^2 - and $\beta^{2,2}$ -amino ketones obtained from the transformations with imines. Product **12b** from imine **22** should come from a Mannich-type radical addition for its higher concentration than the α -amino radical generated from **2a** in the process and the excellent reactivity which has been described by Knowles and co-workers (See below figure) and could be demonstrated by the result that no ketone **24** as the debrominative protonation product was detected (Note: in the reactions as shown in Fig. 4, the debrominated ketone was determined as a side product).

Rono, L. J., Yayla, H. G., Wang, D. Y., Armstrong, M. F. & Knowles, R. R. Enantioselective Photoredox Catalysis Enabled by Proton-Coupled Electron Transfer: Development of an Asymmetric Aza-Pinacol Cyclization. *J. Am. Chem. Soc.* **135**, 17735–17738 (2013)

It is also worth mentioning that no reaction was detected when three kinds of the parent ketones with imine **22** were under the corresponding standard reaction conditions.

In conclusion, a radical coupling process to generate the products could be demonstrated by the investigations as depicted in Fig. 5B and 5C.

We appreciate the referee for the careful consideration.

Sincerely Yours

Zhiyong Jiang Dr.

Reviewer #3 (Remarks to the Author):

The new competition experiments (described in Figure 5c) provide a better rationale for two separate mechanisms. The details should be included in the supplemental materials.

Similarly, the authors should identify the "three kinds of the parent ketones [that did not react] with imine 22" under these conditions.

Point to Point Response to Reviewers' Comments

Reviewer 3' comments and our responses

- 1) The new competition experiments (described in Figure 5c) provide a better rationale for two separate mechanisms. The details should be included in the supplemental materials. Similarly, the authors should identify the "three kinds of the parent ketones [that did not react] with imine 22" under these conditions.

Response: For the experiment described in Figure 5c, the details have been added in page S115 of SI. For the three experiments of the parent ketones with imine 22, the figure and description have been shown in page S115 of SI.

We appreciate the referee for the careful consideration.